# Fully automated sequence alignment methods are comparable to, and much faster than, traditional methods in large data sets: an example with hepatitis B virus

Therese A. Catanach[1,2,3,*], Andrew D. Sweet[2,15,*], Nam-phuong D. Nguyen[5], Rhiannon M. Peery[6,7], Andrew H. Debevec[8], Andrea K. Thomer[9], Amanda C. Owings[4], Bret M. Boyd[2,10], Aron D. Katz[2,11], Felipe N. Soto-Adames[12,13] and Julie M. Allen[14]

[1] Ornithology Department, Academy of Natural Sciences of Drexel University, Philadelphia, PA, United States of America

[2] Illinois Natural History Survey, University of Illinois at Urbana-Champaign, Champaign, IL, United States of America

[3] Department of Wildlife and Fisheries Sciences, Texas A&M University, College Station, TX, United States of America

[4] Program in Ecology, Evolution, and Conservation Biology, University of Illinois at Urbana-Champaign, Urbana, IL, United States of America

[5] Computer Science and Engineering, University of San Diego, California, La Jolla, CA, United States of America

[6] Department of Biology, University of Alberta, Edmonton, Alberta, Canada

[7] Department of Plant Biology, University of Illinois at Urbana-Champaign, Champaign, IL, United States of America

[8] School of Integrative Biology, University of Illinois at Urbana-Champaign, Champaign, IL, United States of America

[9] School of Information, University of Michigan—Ann Arbor, Ann Arbor, MI, United States of America

[10] Department of Entomology, University of Georga, Athens, GA, United States of America

[11] Department of Entomology, University of Illinois at Urbana-Champaign, Champaign, IL, United States of America

[12] Florida State Collection of Arthropods, Florida Department of Agriculture and Consumer Services, Gainesville, FL, United States of America

[13] Department of Entomology and Nematology, University of Florida, Gainesville, FL, United States of America

[14] Biology Department, University of Nevada, Reno, Reno, NV, United States of America

[15] Department of Entomology, Purdue University, West Lafayette, IN, United States of America

[*] These authors contributed equally to this work.

Corresponding author
Therese A. Catanach,
tacatanach@drexel.edu

## ABSTRACT

Aligning sequences for phylogenetic analysis (multiple sequence alignment; MSA) is an important, but increasingly computationally expensive step with the recent surge in DNA sequence data. Much of this sequence data is publicly available, but can be extremely fragmentary (i.e., a combination of full genomes and genomic fragments), which can compound the computational issues related to MSA. Traditionally, alignments are produced with automated algorithms and then checked and/or corrected "by eye" prior to phylogenetic inference. However, this manual curation is inefficient at the data scales required of modern phylogenetics and results in alignments that are not reproducible. Recently, methods have been developed for fully automating alignments

of large data sets, but it is unclear if these methods produce alignments that result in compatible phylogenies when compared to more traditional alignment approaches that combined automated and manual methods. Here we use approximately 33,000 publicly available sequences from the hepatitis B virus (HBV), a globally distributed and rapidly evolving virus, to compare different alignment approaches. Using one data set comprised exclusively of whole genomes and a second that also included sequence fragments, we compared three MSA methods: (1) a purely automated approach using traditional software, (2) an automated approach including by eye manual editing, and (3) more recent fully automated approaches. To understand how these methods affect phylogenetic results, we compared resulting tree topologies based on these different alignment methods using multiple metrics. We further determined if the monophyly of existing HBV genotypes was supported in phylogenies estimated from each alignment type and under different statistical support thresholds. Traditional and fully automated alignments produced similar HBV phylogenies. Although there was variability between branch support thresholds, allowing lower support thresholds tended to result in more differences among trees. Therefore, differences between the trees could be best explained by phylogenetic uncertainty unrelated to the MSA method used. Nevertheless, automated alignment approaches did not require human intervention and were therefore considerably less time-intensive than traditional approaches. Because of this, we conclude that fully automated algorithms for MSA are fully compatible with older methods even in extremely difficult to align data sets. Additionally, we found that most HBV diagnostic genotypes did not correspond to evolutionarily-sound groups, regardless of alignment type and support threshold. This suggests there may be errors in genotype classification in the database or that HBV genotypes may need a revision.

**Subjects** Bioinformatics, Computational Biology, Data Mining and Machine Learning
**Keywords** Genome, Automated alignment, Manual alignment, Virus, s-region, HBV

## INTRODUCTION

The multiple sequence alignment (MSA) is arguably one of the most important steps in a phylogenetic analysis (*Morrison, 2006*; *Kemena & Notredame, 2009*), but can be difficult to perform accurately on large data sets (*Liu, Linder & Warnow, 2010*). Traditionally, MSA methods for phylogenetic reconstruction are performed using alignment programs such as MUSCLE (*Edgar, 2004*) followed by manual, "by eye" corrections (*Hillis, Moritz & Mable, 1996*; *Hall, 2001*; *Edgar & Batzoglou, 2006*). Studies such as *Kjer, Gillespie & Ober (2007)* found that manual curation of MSA data was common in studies published in journals focused on phylogenetics. Google Scholar searches including terminology associated with manual editing of MSA reveals tens of thousands of hits (search completed 21 March 2018). Similarly, *Morrison (2009)* surveyed 247 systematics papers published in 2007 and found that 76% of these studies included a manual curation step. Therefore, while often only mentioned in passing, manual editing of MSA is a common practice. However, as data sets have grown larger, manual examination of each alignment has become increasingly difficult. Furthermore, the results of such effort cannot be replicated and may lead to

inconsistencies between studies. These differences may be significant in data sets where manual editing is especially challenging: for example, data sets with many sequences (>10,000) and varying sequence lengths. This problem can become increasingly difficult if there are repeat elements in the genome and rapidly evolving sites. Although methods have recently been developed for fully automating alignments with large data sets, (e.g., PASTA; *Mirarab et al., 2014*) and for adding in fragmentary sequences to longer stretches of DNA (e.g., MAFFT–addfragments; *Katoh et al., 2002*; *Katoh & Martin, 2012*, UPP; *Nguyen et al., 2015*), it is unclear if these fully automated methods result in equivalent alignments (i.e., result in trees with similar topologies) when compared to an alignment created using the traditional approach.

In recent decades, molecular phylogenetics has experienced a boom due to the generation of increasingly larger DNA sequence data sets for addressing difficult biological questions (*Eisen & Fraser, 2003*; *Delsuc, Brinkmann & Philippe, 2005*; *Philippe et al., 2005*; *Jarvis, 2016*). This increase in data has resulted in many phylogenetic studies with thousands of taxa and characters (e.g., thousands of genes, transcriptomes, etc.; *Kozlov, Aberer & Stamatakis, 2015*; *Arbizu et al., 2014*; *Jarvis et al., 2014*; *Misof et al., 2014*; *Faircloth et al., 2013*; *Heyduk et al., 2015*; *Leache & Linkem, 2015*; *Harkins et al., 2016*). While larger data sets may help resolve difficult phylogenetic questions, they also present computational challenges (*Sanderson & Driskell, 2003*; *Soltis, Gitzendanner & Soltis, 2007*). A byproduct of the increased DNA sequencing efforts is the large amount of publicly available sequences in databases including GenBank, Ensemble, and DDBJ (*Benson et al., 2014*; *Yates et al., 2016*; *Mashima et al., 2016*). Although the number of whole genome sequences in these databases will continue to increase with the widespread adoption of next generation sequencing (NGS) technology, currently a considerable portion of available data are genomic fragments (e.g., from Sanger sequencing; (*Benson et al., 2014*; http://www.ncbi.nlm.nih.gov/genbank/statistics/).

Viruses can pose particularly unique challenges for MSA and phylogenetic inference. These organisms often have relatively small genomes and undergo rapid evolution, making it difficult to assess homology within alignments (*Beerenwinkel et al., 2012*; *Nasir & Caetano-Anolles, 2015*). However, due to their medical relevance, viruses are well-represented in public sequence databases (*Chooka et al., 2015*; https://www.ncbi.nlm.nih.gov/genome/viruses/) as a mix of full genomes and sequence fragments. In most cases, available sequences from these taxa are from regions of the genome selected for diagnostic purposes (*Weber, 2005*). These areas tend to be fast evolving and short in length (*Osiowy et al., 2006*). Incorporation of these sequences in phylogenetic analyses is useful, but integrating them into a MSA with full genome sequences can be challenging. Global alignments find similarities between sequences from beginning to end and local alignments try to find a similar region to align from in all sequences (*Philippe et al., 2011*; *Nguyen et al., 2015*). Thus, both approaches fail completely when sequences have no characters in common. New approaches that are a hybrid of alignment methods, an increasingly active research area, are appropriate for data with mixed fragment and genome information (*Schmollinger et al., 2004*; *Subramanian et al., 2005*; *Liu et al., 2012*; *Hossain et al., 2013*; *Ye et al., 2015*; *Nguyen et al., 2015*).

Here we focus on the hepatitis B virus (HBV), a medically important and globally distributed virus (*Okamoto et al., 1988*; *Norder, Courouce & Magnius, 1994*; *Stuyver et al., 2000*; *Arauz-Ruiz et al., 2002*; *Simmonds & Midgley, 2005*; *Tran, Trinh & Abe, 2008*; *Tatematsu et al., 2009*; *Kurbanov, Tanaka & Mizokami, 2010*). Every year ca. 750,000 people die from the virus (*Lozano et al., 2012*) and in some geographic regions up to 20% of the adult population have chronic HBV (*Chen, 1993*; *Liaw & Chu, 2009*). There are currently 10 recognized genotypes of HBV designated A-J. These genotypes are primarily associated with particular geographic regions, but are also identified based on genetic divergence (*Hernández et al., 2014*). A push to understand the global distribution and diversity of the virus (*Shi et al., 2013*), along with associated global health implications (*Shi, 2012*), has resulted in large amounts of publicly available HBV sequence data. The S gene region (barcoding region of the HBV genome; (*Galibert et al., 1979*)), is the most abundant HBV sequence in GenBank. However, many full genome sequences are also available (*Wu, Ding & Zeng, 2008*).

HBV has a compact circular genome ∼3.2 kb in length, with four genes arranged across overlapping reading frames (Fig. 1; *Norder, Courouce & Magnius, 1994*). Any attempt to use publicly available HBV sequence data for phylogenetic analysis needs to work with the fragmentary nature of the data set and with the issues of limited character sampling (i.e., small genomes). The small size of the genomes coupled with the large number of sequences creates phylogenetic data sets that are "tall and narrow". Furthermore, there is currently no universally agreed starting position for viruses with circular genomes, including HBV, therefore individual researchers are free to report genome sequences starting at any position, and genomes may have arbitrary breakpoints. This results in sequence alignment issues as homologous portions of the genome are not necessarily in the same relative position when represented as linear sequences.

To ameliorate these issues in virus MSA, we compared traditional automated-manual alignment approaches to fully automated methods for phylogenetic analyses on a tall and narrow data set generated from publicly available HBV sequence data. Using two different data sets, one comprised exclusively of whole genomes (and therefore with limited missing data) and a second comprised of most publicly available HBV sequences (a highly fragmentary data set) we compared three alignment methods: (1) traditional automated MSA methods , (2) the commonly used hybrid approach of MSA followed by manual adjustment of the alignment, and (3) UPP and PASTA, two modern automated MSA methods. This approach allowed us to assess the effectiveness of different methods for handling the problem of large matrices with highly fragmentary sequence data. Additionally, through MSA evaluation methods we were able to assess the phylogenetic signal of and monophyly within named strains of the virus.

## METHODS

### Data downloads and initial cleaning

To obtain sequence and genotype data we downloaded all GenBank files (.gb) for hepatitis B sequences using the NCBI Taxonomy Browser (Taxid: 10407) on 20 March 2013. This

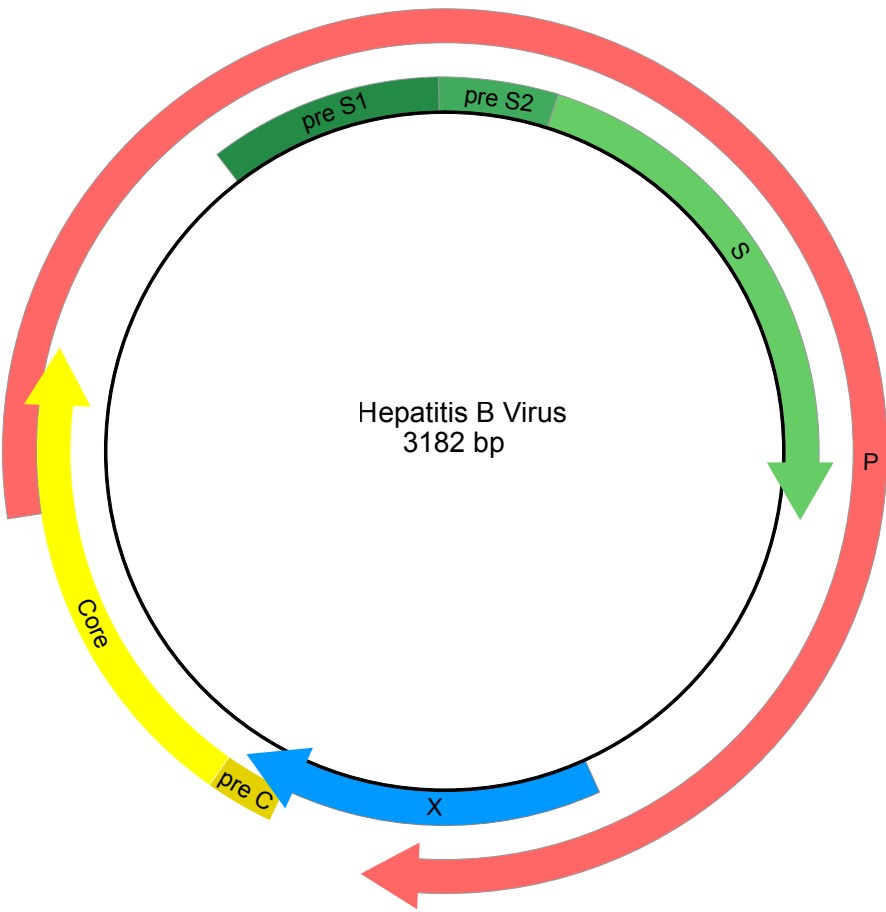

**Figure 1** **Genetic map of the hepatitis B virus genome.** Arrows indicate the reading frames of genes.

included both full genome and fragmentary sequence data. This initial download yielded 55,353 sequences. Sequences from non-primate hosts, less than 100 bp in length, lab strains, and recombinant genomes were removed.

## Sequence alignment

Here we detail our approaches for aligning genomes only and all (genomes + fragmentary) HBV sequence data sets (Table 1). The workflows for both approaches are summarized in Fig. 2 and a more detailed version can be found in Fig. S1. Commands for relevant programs are listed in the supplementary information (see Supplementary Information on Dryad: https://doi.org/10.5061/dryad.nc220).

## Genome alignments
### *Traditional alignment*

To generate the genomes-only data set, we selected whole genomes from our initial filtered GenBank download. Whole genomes were identified by sequence length. The HBV genome is estimated to be ca. 3.2 kb long, therefore we selected sequences ≥3,000 bp to be included in our full genome data set. In total, we identified 5,553 full genome sequences.

**Table 1 Comparison of different alignment approaches used for the genome and total (genomes + fragments) data sets from HBV.**

| Method | Genomes only | All sequences |
|---|---|---|
| Traditional | MUSCLE + manual | MUSCLE + manual |
| Automated | Multiple methods (See Table 2) | UPP: Genomes-manual backbone |
| Automated | PASTA | UPP: PASTA backbone |

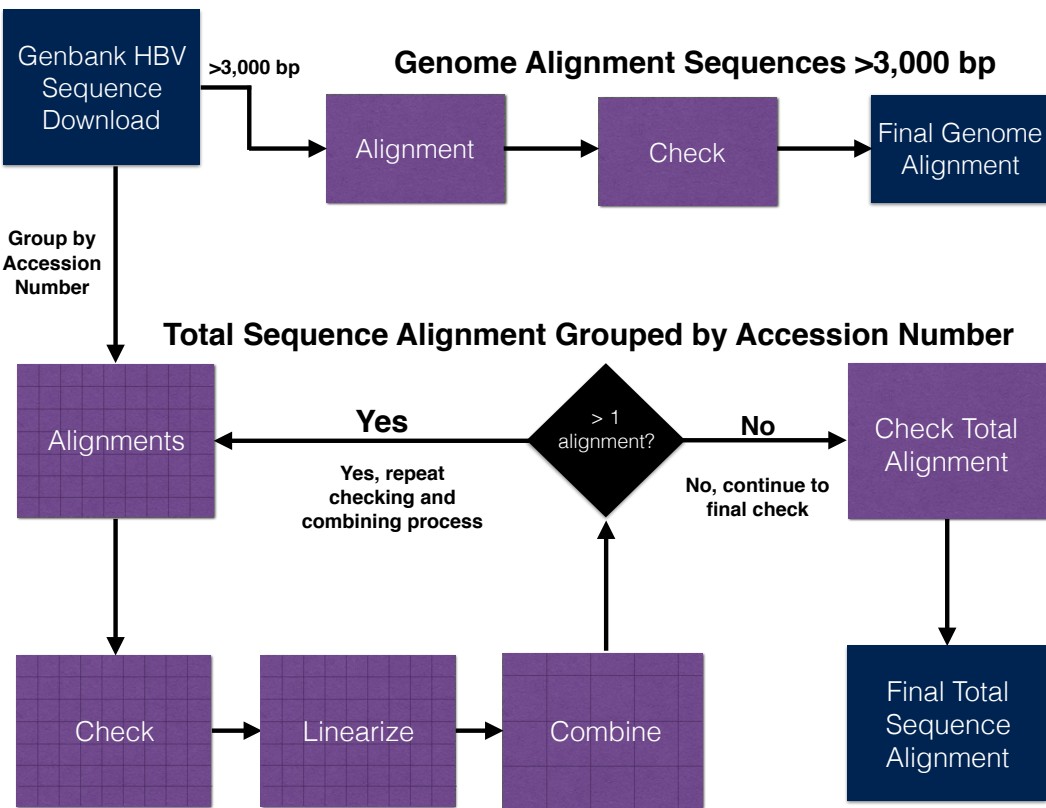

**Figure 2 Workflow outline for the alignment of hepatitis B virus genome and total (genomes + fragments) data set.** Additional details for each step are illustrated in Fig. S1.

There is not a universal starting point within the HBV genome, necessitating the development of a customized script to "linearize" the HBV sequences for the alignment (https://github.com/tacatanach/Hepatitis_B_Virus_alignment). This script chose a universal starting point in the alignments, and removed any overhang from the end and placed it at the beginning. The "linearized" sequences were then aligned using the default gap parameters in MUSCLE (*Edgar, 2004*) and manually adjusted using SeaView v.4 (*Gouy, Guindon & Gascuel, 2010*). We denote this alignment as the "MUSCLE-manual alignment". In addition, we aligned the genomes without manual adjustment using MUSCLE, Clustal-Omega, MAFFT, and PASTA (see Table 2 for version information). However, we primarily focused on the MUSCLE-manual alignment to be comparable with

**Table 2 Comparison of alignment methods.** All alignments were attempted using Genomes_Manual_degapped.fasta as the input file.

| Method | Alignment length | Wall clock time | Method reference |
| --- | --- | --- | --- |
| Manual | 4,269 | | NA |
| PASTA | 3,423 | 3.5 h | *Mirarab et al. (2014)* |
| MAFFT | 4,578 | 4 min | Mafft 7.305b; *Katoh & Standley (2013)* |
| MUSCLE | 4,938 | 32 h | Muscle v3.8.31; *Edgar (2004)* |
| Clustal Omega | 3,846 | 2.25 h | Clustal Omega 1.2.4; *Sievers et al. (2011)* |

analyses using the total data set (see below). Sequences that were not alignable because they were either too taxonomically divergent, had reported recombination between strains in the GenBank information, or had evidence of unreported recombination were removed. Unreported recombination was inferred when a region did not align well and BLAST results indicated that the region was a strong match for a different region of the reference genome. These recombinants can be the result of both inter- and intra-strain recombination of the genome. This alignment was manually checked by three authors.

As a final quality check of the MUSCLE-manual alignment, we inferred a neighbor-joining (NJ) tree using the R package *ape* (*Paradis, Claude & Strimmer, 2004*). In order to remove poorly aligned sequences, we iteratively ran NJ quality control steps (NJ QC; available at https://github.com/tacatanach/Hepatitis_B_Virus_alignment) as follows. We flagged sequences with outlier branch lengths (branches whose leaf to the nearest node distance were >1.5 times the interquartile range of the other branch lengths), and removed these outliers from the alignment. We then repeated the NJ QC step two more times to remove any remaining poorly aligned sequences. The final alignment included 5,244 sequences.

### Automated alignment

To compare the traditional method of MSA (using alignment software followed by manual adjustment) with fully automated approaches, we started with the 5,244 individual sequences identified in the manual method as high quality genomes. We then aligned these sequences with several alignment methods without manually adjusting the resulting alignment. The input data were linearized and degapped prior to alignment. For alignment software we used PASTA v.1.6.3 (*Mirarab et al., 2015*), MAFFT v7.305b (*Katoh & Standley, 2013*), Clustal Omega v1.2.4 (*Sievers et al., 2011*), and Muscle v3.8.31 (*Edgar, 2004*). In addition, we used PASTA to align the original 5,244 (i.e., un-linearized) genome sequences.

### Total (fragments + genome) alignments
### Traditional alignment

A traditional, brute force approach to align all 55,353 sequences (including full genomes and fragmentary sequences) was not feasible due to the large number of sequences and alignment gaps from the fragments. Therefore, to align the full set of sequences we first used a custom script to group sequences into 1,269 FASTA files based on their GenBank accession numbers (https://github.com/tacatanach/Hepatitis_B_Virus_alignment). Because GenBank accession numbers are assigned sequentially at submission, a cluster of successive

HBV sequences is likely from the same region of the genome. As with the genome sequences, we used our custom script to "linearize" the original GenBank sequences. We also created a consensus sequence of the full HBV genome (3,639 bp) based on the genome alignment, using majority rule (rather than ambiguity codes) for ambiguous sites. This consensus sequence served as a reference for downstream alignment steps. Each author then aligned about 150 files, using the default gap parameters in MUSCLE and checking each alignment by eye against the genome consensus sequence. To facilitate aligning by eye we used custom scripts to "ladderize" the alignments by ordering the sequences according to starting position and length (https://github.com/tacatanach/Hepatitis_B_Virus_alignment) so that in each alignment sequences with similar starting points and lengths were placed next to each other. To combine two alignments we used profile-profile in MUSCLE followed by manual adjustments against the genome consensus. We iteratively repeated this process to gradually combine the sequences. This process was successful for several rounds of profile-profile alignments. However, the alignment files eventually became too large for the profile-profile function in MUSCLE to consistently align pairs of files. At this point, we began manually combining pairs of aligned sequence files using the *cat* Unix command followed by manual adjustments. Adjustments were made by opening gaps in each alignment subset to align with the reference.

Additionally, there were 930 individually uploaded sequences ("singletons"; i.e., sequences that did not cluster with other sequences according to accession number). To align these sequences with the larger sequence matrix, we first split all singletons into files of 50–100 sequences, each file containing a consensus genome sequence. These sequences were then aligned by eye to the consensus genome sequence, and resulting aligned sequence files were combined as described above. We then manually combined all aligned sequence files as described above, using the genome consensus sequence as a reference.

To further check the alignment, we created consensus sequences (with ambiguity codes) for blocks of 73 (the number of lines viewable in a screen without scrolling; for ease of viewing) sequences (https://github.com/tacatanach/Hepatitis_B_Virus_alignment). We then combined all consensus sequences maintaining the alignment structure. By collapsing sets of sequences into a single consensus sequence we could visually check the entire alignment. If consensus sequences were not aligned to each other or contained many ambiguity codes, this indicated a particular portion of the alignment was misaligned. We then adjusted the corresponding alignment accordingly and repeated the process until we could not detect any misaligned regions.

During the alignment step, 5,000 sequences could not be aligned because they were unflagged recombinants or from non-humans hosts and were subsequently removed. We also ran five iterations of the NJ QC step and removed an additional 168 sequences. Finally, we checked to ensure that sequences from unique hosts, geographic regions, ancient (more than 100 years old) samples, and all genotypes were present in the alignment. If any of these key data were removed through prior filtering steps, they were manually added back into the alignment. Including these sequences is important for having a representative sample, and warranted the additional effort required to incorporate them into our alignment. We

also included a sequence of HBV from woolly monkey (GenBank accession JX978431) as an outgroup.

### Automated alignment

In order to obtain an automated alignment of the total data set, we used UPP v.3.0 (*Nguyen et al., 2015*), a method designed for inserting fragmentary sequences into an existing backbone alignment using a backbone tree to guide the process. To evaluate the influence a backbone tree may have on UPP alignments, we ran a pair of UPP analyses, one with a tree from the PASTA genomes-only alignment as a backbone and a second using a tree from the MUSCLE-manual genomes-only alignment. We estimated Maximum Likelihood (ML) backbone trees under GTR+CAT using FastTree-2 v.2.1.7 (*Price, Dehal & Arkin, 2010*) on the backbone alignments. We selected PASTA as the automated alignment to compare against the MUSCLE-manual alignment because PASTA uses MAFFT internally and results in a similar backbone tree topology as the Clustal-Omega and MAFFT alignments (Fig. S2).

### S-region

The HBV S-region is often sequenced for identification of viral genotype (*Galibert et al., 1979*). Therefore, as a comparison to the genomes-only and total (genomes + fragments) data sets, we identified and analyzed the S-region in the whole manual genome alignment based on the annotation of GenBank accession AJ131956 (*Ozaslan et al., 2007*). The annotation of the S-region combines pre-S1, pre-S2, and the S gene.

## Tree estimation

For each data set we inferred a ML phylogeny with bootstrap support. First, we generated 100 bootstrap replicate alignments in RAxML v.8.2.10 (*Stamatakis, 2014*). For each bootstrap replicate we estimated an ML tree using FastTree2. We then used FastTree to estimate a best ML tree from the original alignment, and summarized the bootstrap replicates on this tree using SumTrees v. 4.0.0 in DendroPy (*Sukumaran & Holder, 2010*).

## Tree comparisons

To compare the similarity of tree topologies, we did pairwise comparisons of the trees produced using each of the MSA strategies. For each of the trees we collapsed nodes based on three bootstrap support thresholds (50%, 75%, 90%). We then calculated the total number of edges for each tree and the number of edges that were incompatible with the other tree in the pairwise comparison using Phylonet Modified (*Mirarab et al., 2014*). We also calculated Robinson-Foulds (RF) distances (*Robinson & Foulds, 1981*) for each tree comparison using CompareTree.pl (*Fast Tree-Comparison Tools, 2009*). However, due to how RF distances are calculated, comparing this metric across trees can be misleading. For example, two clades which are identical in topology except for the placement of one taxon can have a high RF value if the one non-identical node is placed at the tip of a clade in one tree but at the base in the second tree. Therefore, although we include the RF metric, we interpret our results based primarily on the incompatibility ratios. We performed all possible pairwise comparisons between the different trees for a specific support cutoff. We then converted each comparison to a ratio of the number of incompatible edges to the total

number of edges in the comparison. For comparisons between genome and total alignment trees, we pruned the total alignment trees to be the same set of taxa as the genome-only alignments.

As a way to assess tree differences in a biologically meaningful context, we mapped genotype labels onto our tree tips. We pruned out tips that did not have any associated genotype data or were recombinant (i.e., were associated with multiple genotypes) based on GenBank metadata. For each genotype, we then identified the smallest possible clade that included all taxa of the genotype, and calculated the percentage of tips in the clade identified as that genotype using a python script mono.py (https://github.com/tacatanach/Hepatitis_B_Virus_alignment). We call this metric the "genotype occupancy proportion." We repeated this analysis for each bootstrap collapse threshold (50%, 75%, and 90%) of each tree.

# RESULTS

## Data sets

### Full genomes and S-region

The initial genomes-only alignment was 7,069 characters long. Quality trimming steps for the genomes-only data set removed 309 sequences. "Linearizing" and manually aligning the trimmed subset resulted in a sequence matrix with 5,244 taxa and 4,269 characters. The MUSCLE alignment with the same sequences resulted in an alignment with 4,661 characters, and the PASTA alignment was 3,423 characters long. The S-region-only alignment from the manual genomes alignment was 1,340 characters long. All three alignments reveal similar pairwise sequence dissimilarity distributions Fig. S3, Table S2 on Dryad), with a mean $p$-distance of 0.08 and max $p$-distance of 0.20.

### Total data set

All sequences included in the genome-only alignment were also included in the total alignment. After filtering the initial NCBI download, we had a data set of 37,788 sequences, including both full genome and fragmentary sequences (Table S1 on Dryad, available at https://doi.org/10.5061/dryad.nc220). After iteratively aligning sequence files and filtering out divergent sequences (recombinants, etc.), the final alignment included 32,819 sequences. "Linearizing" the full alignment and removing gap-only columns resulted in a final alignment length of 5,196 characters. This same data set was used to create two UPP alignments: (1) using the manually-aligned genomes only data set as a backbone (final alignment: 4,205 characters); and (2) using a PASTA-aligned backbone (final alignment: 3,387 characters).

### Tree comparisons

Comparisons between trees from the different alignment types showed different levels of compatibility depending on bootstrap cutoff and alignment type, and are summarized in Figs. 3 and 4. Specific values are listed in Table S1 (on Dryad). Trees without any edges collapsed (0% threshold) had the highest incompatibility ratios. The lowest ratio values for a 0% threshold were from comparisons between genome alignment trees, but these

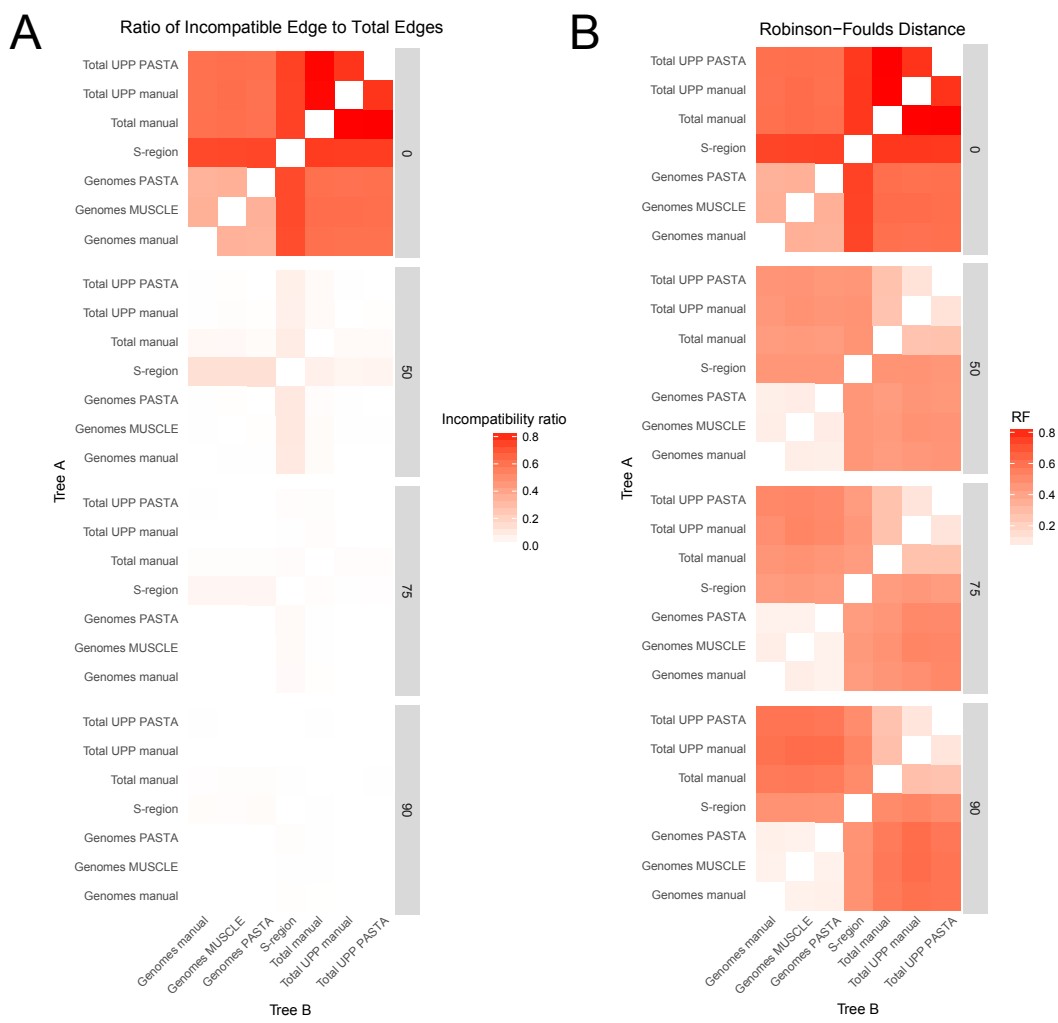

**Figure 3** **Heatmaps representing the pairwise comparisons of hepatitis B virus phylogenies.** (A) Ratio of incompatible edges to total edges in a comparison. (B) Robinson–Foulds distances between two phylogenies. Darker cells indicate two trees with greater differences. Alignments used to estimate the phylogenies are indicated on the *x*- and *y*-axes.

were still considerably higher than ratio values from comparisons at all other thresholds. Incompatibility ratios tended to decrease with an increasing bootstrap threshold. The lowest ratio values were for comparisons at the 75% and 90% thresholds. Ratios for comparisons at the 50% threshold tended to be higher than those for comparisons at the 75% and 90% thresholds, but were all considerably lower than comparisons at the 0% threshold. RF values also tended to be higher for comparisons at the 0% threshold, but this was not true across all comparisons. The lowest RF values were from comparisons at higher thresholds. However, there was not a clear pattern among comparisons at the 50%, 75%, and 90% thresholds. For example, several 90% threshold comparisons had higher RF values than some 50% and 75% threshold comparisons.

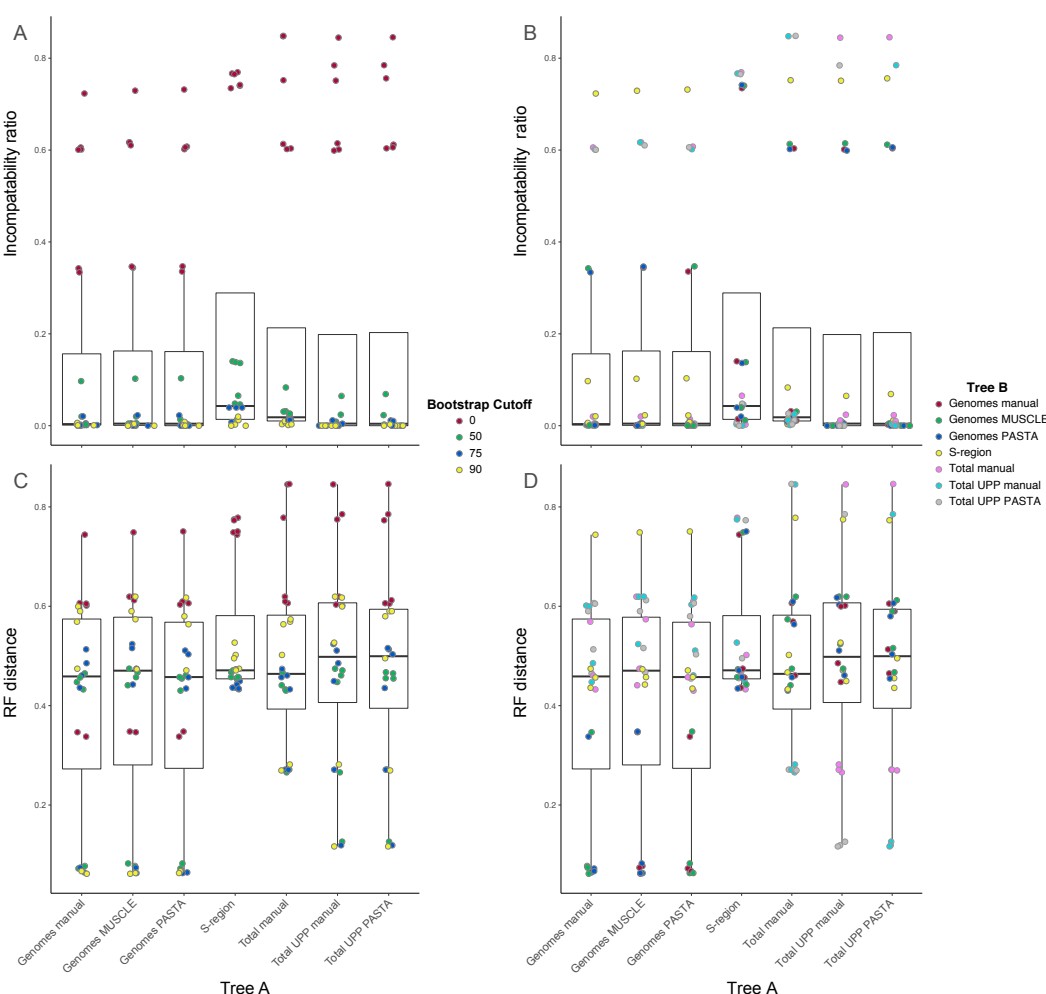

**Figure 4** **Box-and-whisker plots showing the distributions of pairwise comparison values for hepatitis B virus phylogenies.** (A) and (B) show the ratio of incompatible edges to total edges in a comparison. (C) and (D) show Robinson-Foulds distances. Alignments used to estimate the phylogenies are indicated on the *x*-axis. (A) and (C) are colored according to the edge collapse threshold (bootstrap value). (B) and (D) are colored according to the second tree in a comparison.

Among trees from the different alignment types, trees from the S-region alignment tended to have higher incompatibility ratio values. This was true at each bootstrap threshold. Comparisons including genome alignment trees tended to have the lowest ratios whereas the highest values involved comparisons between the genome and total alignments. Trees from total (genomes + fragments) alignments had intermediate ratios. Trees from the total automated alignments (with UPP) had slightly lower ratios than the total manual alignment, but the two UPP trees had the fewest number of differences between them (Fig. 4, Table S1 on Dryad). For the RF metric, the lowest values came from comparisons among major alignment types. Genome-genome comparisons and total-total comparisons had the lowest RF values. Trees from the S-region alignment had similar RF values across comparisons with other trees, except for comparisons at the 0% threshold.

The average absolute difference in genotype occupancy proportions between the different alignments was 4.6% for genomes and S-region trees, and 2.1% for total alignment trees. Overall, the average proportions of genotype occupancy across all alignment types, genotypes, and support thresholds were 52% for genome and S-region trees, and 13.1% for total alignment trees. These patterns are summarized in Fig. 5, and specific genotype occupancy proportion values are listed in Table S2 (on Dryad). However, there was a great deal of variability between genotypes. Genotypes A–C, G, and genotype I consistently exhibited clade occupancy values between 15%–38% among the different alignment types (genomes, S-region, total) and bootstrap support collapse cut offs. Conversely, genotypes E, F, and H consistently accounted for most (85%–100%) of the tips in their respective clades in genome and S-region trees, but lower than 10% in the total alignment trees. Although these genotypes are relatively rare in GenBank (150 or fewer genome sequence and 530 or fewer total sequences per genotype), they are geographically widespread (*Shi et al., 2013*). The addition of fragmentary sequence data could have greatly increased the geographic coverage of the data set which could result in an increase in genetic variation within the genotype and an extreme decrease in genotype occupancy. Additionally, as we were limited to GenBank annotations for identifying genotype, it is possible that the inclusion of the fragmentary data, an inclusion that in some instances increased the number of sequences six-fold, resulted in introducing sequences that had been incorrectly annotated.

Genotype D showed more complex patterns than the other genotypes. On one hand, genotype D accounted for a high proportion (99%) of tips in the "D" clade for most genome alignment types and support thresholds. However, the S-region only illustrated a high proportion of genotype occupancy at the 0% bootstrap cutoff. Additionally, at the 90% bootstrap cutoff only the manual genome alignment had a high occupancy value, whereas the other two genome alignments (MUSCLE and PASTA) had a value of 22%. This lower value was similar to occupancy proportion values in all total alignment categories for genotype D.

### Labor requirements

We compare the labor requirements for the automated alignment methods (MUSCLE, Clustal Omega, MAFFT, PASTA) with the fully manual MUSCLE-manual alignment. All automated alignments were run on a dedicated machine with 24 nodes. Automated methods that were parallelized completed quickly, with MAFFT requiring 4 min of wall clock time, Clustal Omega 2.25 h, and PASTA requiring 3.5 h. MUSCLE, on the other hand, is a single threaded program and required 32 h to align. An additional 72 person-hours were needed to manually hand-curate the MUSCLE alignment to produce the MUSCLE-manual alignment. It is important to note that both PASTA and UPP can result in slightly differing alignments over multiple runs due to the use of multiple cores.

An additional 36 h were needed to align the total data set (27,575 fragments plus whole genomes) using UPP and the genome backbones (from PASTA and the manual alignment). Although an exact estimate of the manual alignment of this same data set is not possible, in comparison the process took >1,000 person-hours divided between nine authors over a two year period.

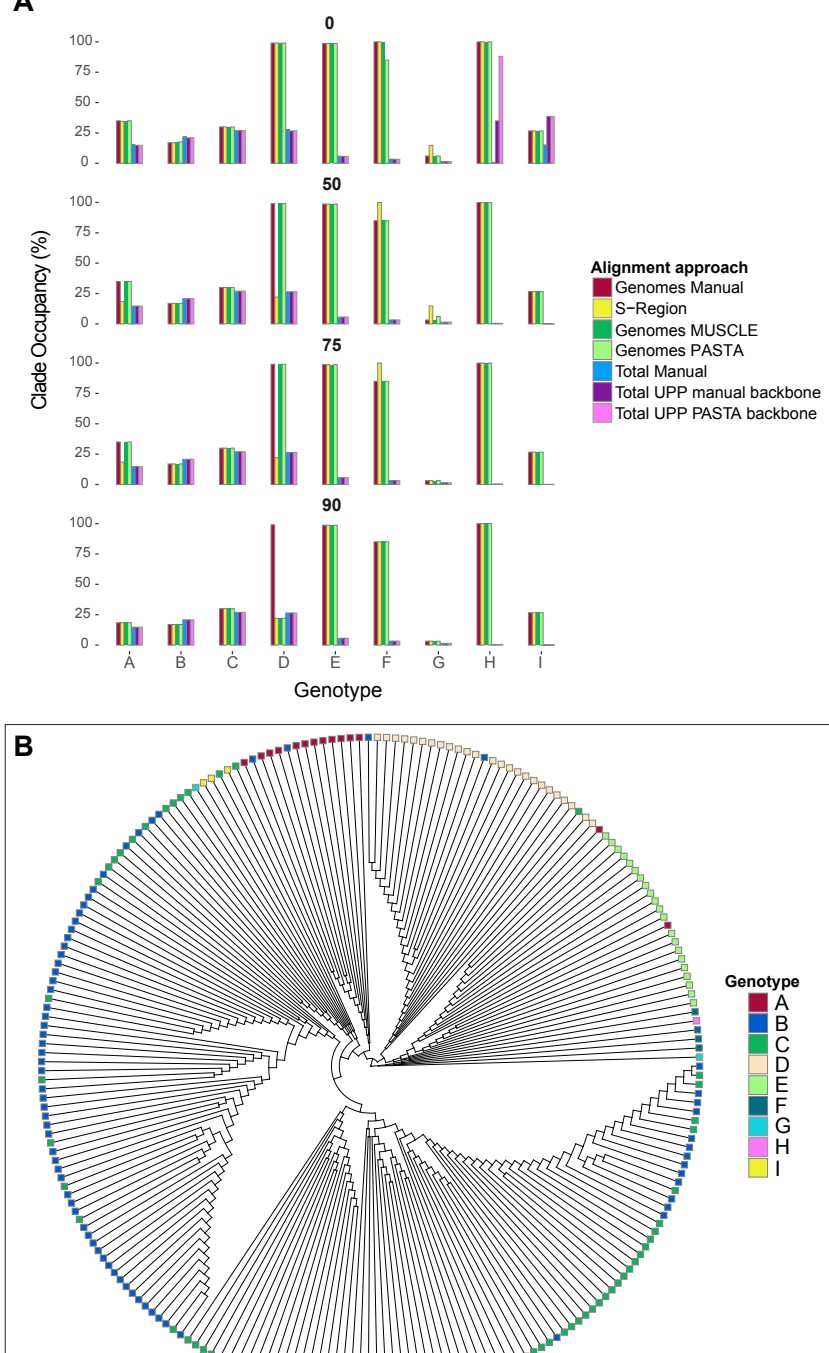

**Figure 5** **Hepatitis B virus genotype occupancy.** (A) Histograms showing the proportion of each genotype that makes up the minimum clade including all individuals of that genotype. Genotypes are indicated on the *x*-axis, and genotype occupancy is shown as percentages along the *y*-axis. Bootstrap support collapse thresholds are indicated above each panel. (B) Fully bifurcating (i.e., no branches collapsed), midpoint-rooted cladogram of HBV sequences from the PASTA genome alignment. Tips are colored according to genotype.

## DISCUSSION

Comparisons among phylogenetic trees of HBV indicated that sequence alignment strategy did not have a major effect on tree topology. Within the data set containing only complete genomes, all alignment methods (ClustalOmega, PASTA, MAFFT, MUSCLE, MUSCLE with manual adjustments, or completely manual) produced phylogenies without noticeable differences using three methods of evaluation (clade occupancy, RF values, and incompatibility ratios). Additionally, when two of these whole genome alignments (PASTA and manual alignments) served as a backbone to align genome fragments with UPP, the resulting phylogenies were not different in any biologically meaningful way based on our metrics.

When considering support values for phylogenetic trees produced by different alignment methods, trees with a minimum acceptable bootstrap support threshold of 50% or greater were compatible (less than 20% of edges were incompatible). These results suggest multiple alignment approaches are effective for highly fragmented data sets. However, because manual editing of large fragmentary data sets is laborious, the fully automated approach using appropriate software (e.g., UPP or MAFFT—addfragments) to align these types of data sets is preferred. The traditional method of checking automated alignments consumes time that could be devoted to other tasks, can be difficult to replicate, and is impractical for large data sets. Yet accuracy remains the goal of phylogenetic estimation and is dependent on the alignment step and is the driver for retention of manual curation. Here we have demonstrated that manual alignment and curation give comparable results to automated methods. This is encouraging, given that publically available data from high-throughput NGS sequencing projects is increasing exponentially and will likely drive a greater reliance on automated methods. It also suggests that findings based on completely automated methods are comparable to results from earlier studies that may have included a manual step. Integrating complete data with existing fragmentary sequences is crucial for addressing biologically relevant questions and in our data, we have shown this can be done so in a consistent manner across commonly used methodologies.

Although minor differences in tree topologies are expected when multiple analytical approaches are used, these differences may not be biologically meaningful (i.e rearrangement of branches within a clade). To test the impact of alignment type on the biological meaning of the resulting phylogenies we used HBV genotypes, a feature often annotated by researchers when submitting HBV sequences to GenBank. We sought to determine if HBV genotypes where stable in their location in the tree (consistent) and if they were monophyletic (accurate with regard to evolutionary relatedness) within our tree. Our results are consistent that the genotype occupancy in a clade remain stable across alignment types for both genome and fragmentary data sets. This indicates the alignment approach for these types of data sets does not have a major effect on the biological interpretation of the inferred topologies, particularly when testing the monophyly of a particular subset of tips. This is consistent with our test of incompatible edges among topologies when considering bootstrap support. This assessment also supports using the
fully automated approaches, because using considerably more time to manually adjust alignments does not appear to result in different biological interpretations.

Despite genotypes exhibiting consistent clade composition across the different alignment approaches tested, the genome and fragmentary topologies generally had very different occupancy values between them. Topologies from the genome alignments tended to have higher genotype occupancy than topologies from the fragmentary alignments. However, neither data set consistently showed high occupancy. The genotypes that did have good occupancy values (>85%) were often represented by few tips (<100). This pattern of low occupancy was often consistent across support thresholds, suggesting that tree estimation error is not a primary source of the non-monophyly of genotypes. Therefore, most HBV genotypes appear to be paraphyletic or polyphyletic and do not reflect natural groups. The ten HBV genotypes are generally delimited geographically (*Stuyver et al., 2000*), but are also determined based on sequence divergence (*Hernández et al., 2014*). The S-region, which encodes the surface antigen of the virus, is often used as a barcoding region to identify samples to genotypes (Fig. 1; *Stuyver et al., 2000*). In our analysis, the topology from the S-region had similar genotype occupancy values to other full genome alignment topologies, but the values were still low on average. However, the D genotype exhibited notable discrepancies in occupancy values between the S-region and genome alignments, particularly at higher bootstrap collapse thresholds. Additionally, the S-region topology had the highest ratios of incompatible edges with other topologies, regardless of support threshold cutoff. Convergent molecular evolution or horizontal gene transfer could potentially account for discrepancies among trees and genotypes, although our data set is not ideal for demonstrating these mechanisms. Regardless, our results suggest the S-region is not ideal as a basis for phylogenetic interpretation, and that there is a conflict in phylogenetic signal between the S-region and the rest of the genome.

Our HBV DNA alignment is "tall and narrow," characterized by having many individual sequences but relatively few characters. In this instance, the number of sequences in the alignment matrix (height) is much greater than the number of characters (length), particularly for the data set including both genomes and fragments. This total data set has 32,819 sequences (height) and each contained at most 5,196 characters (width) including gap characters. One way to improve phylogenetic resolution of "tall and narrow" data sets is to increase the width of the alignment matrix by adding characters (*Rokas et al., 2003*; *Rokas & Carroll, 2005*; *Hedtke, Townsend & Hillis, 2006*; *Heath, Hedtke & Hillis, 2008*). However, this may not be possible for organisms with small genomes. HBV has a genome size of only approximately 3.2 kbp, and our alignments represent the maximum possible number of characters available. These types of data sets are only going to increase in height. Many of these sequences are also likely to be fragments of the full genome, particularly if data is added from publicly available databases. These characteristics can render traditional alignment approaches unfeasible. Furthermore, many of the organisms that are well-represented in public sequence databases and tend to have smaller genomes are likely to be medically important viruses and bacteria. Accurate phylogenetic estimation of these organisms is necessary for properly understanding their global diversity patterns, a crucial part of epidemiological studies. The results from our study with HBV show that

"tall and narrow" data sets can be rapidly aligned using available software, and that the resulting phylogenies are comparable to phylogenies estimated from sequences aligned with a traditional approach. There were also no substantial differences of biological relevance, in this case monophyly of HBV genotypes. Together these results suggest the rapid, automated approach will be useful for other "tall and narrow" data sets, including those of medically important taxa.

Another issue facing phylogenetic estimation of HBV is the lack of a universal sequence starting point. Although the genomes are circular, sequences are uploaded to GenBank in linear format. Without a universal starting location, publicly available HBV sequences begin at different places in the genome. Additionally, the 3′ end of one sequence may be orthologous to the 5′ end of another sequence. In a sequence alignment, the two sequences would only overlap over a fraction of their entire length, and the whole alignment would be much longer than necessary. Indeed, when we initially used PASTA to align HBV genomes downloaded from GenBank, the alignment was approximately 19,000 bp in length, or 6 times longer than the length of the HBV genome. It was therefore necessary to "linearize" the alignments by moving 3′ sequences to the 5′ end before aligning the sequences using a fully automated approach. A similar approach should be taken by future studies focusing on HBV phylogenetics, or for other organisms with similar data issues. Our script chooses an arbitrary cutoff point for "linearization," but perhaps future approaches should choose alignment starting points based on genes or other biologically relevant information. Databases could also implement a standardized starting point for data deposited from these circular genomes.

## CONCLUSION

Here we demonstrate that UPP and PASTA, completely automated MSA approaches, resulted in alignments that produced phylogenetic trees that are topologically and biologically similar to those produced using a sequential approach that uses manual interventions. However, the fully automated approaches were completed quickly as opposed to the numerous hours spent on adjusting alignments for the traditional method. Although we used PASTA to generate the backbone for aligning the fragmentary data set with UPP, the results of our clade occupancy comparisons between different alignment methods indicate that alignments from other software will produce similar phylogenetic hypotheses for HBV. Researchers are encouraged to test multiple alignment programs before choosing a backbone in which to align fragmentary data against. Finally, phylogenetic trees estimated from both fully automated and traditional alignments indicated most HBV genotypes are not monophyletic. This result suggests traditional HBV genotype delineations should be reevaluated, and should encourage future studies to further address this issue.

## ACKNOWLEDGEMENTS

We are grateful for Patrick Grady, Brendan Morris, Massimo Pessino, and Pranjal Vachaspati for their comments and contributions to this project. We also thank the

University of Illinois Systematics Group and in particular Sydney Cameron for helpful discussions and suggestions, and the staff at Legends, Champaign, IL for technical support.

### Funding
This work was supported by the National Science Foundation (DEB-1239788 and DEB-1342604), which paid for some computational resources and the salaries of Andrew D. Sweet, Bret M. Boyd, and Julie M. Allen. Computational support was provided by the Extreme Science and Engineering Discovery Environment [TG-ASC160042] grant to Nam-phuong D. Nguyen. There was no additional external funding received for this study. The funders had no role in study design, data collection and analysis, decision to publish, or preparation of the manuscript.

### Grant Disclosures
The following grant information was disclosed by the authors:
National Science Foundation: DEB-1239788, DEB-1342604.
Extreme Science and Engineering Discovery Environment: TG-ASC160042.

### Competing Interests
The authors declare there are no competing interests.

### Author Contributions
- Therese A. Catanach, Andrew D. Sweet, Nam-phuong D. Nguyen and Julie M. Allen conceived and designed the experiments, performed the experiments, analyzed the data, contributed reagents/materials/analysis tools, prepared figures and/or tables, authored or reviewed drafts of the paper, approved the final draft.
- Rhiannon M. Peery conceived and designed the experiments, performed the experiments, analyzed the data, prepared figures and/or tables, authored or reviewed drafts of the paper, approved the final draft.
- Andrew H. Debevec conceived and designed the experiments, performed the experiments, contributed reagents/materials/analysis tools, authored or reviewed drafts of the paper, approved the final draft.
- Andrea K. Thomer conceived and designed the experiments, performed the experiments, prepared figures and/or tables, authored or reviewed drafts of the paper, approved the final draft.
- Amanda C. Owings, Aron D. Katz and Felipe N. Soto-Adames conceived and designed the experiments, performed the experiments, authored or reviewed drafts of the paper, approved the final draft.
- Bret M. Boyd conceived and designed the experiments, performed the experiments, analyzed the data, contributed reagents/materials/analysis tools, authored or reviewed drafts of the paper, approved the final draft.

## Data Availability

GitHub: https://github.com/tacatanach/Hepatitis_B_Virus_alignment.

Catanach TA, Sweet AD, Nguyen ND, Peery R, Debevec AH, Thomer AK, Owings AC, Boyd BM, Katz AD, Soto-Adames FN, Allen JM. Data from: Fully automated sequence alignment methods are comparable to, and much faster than, traditional methods in large data sets: an example with hepatitis B virus. Dryad Digital Repository. https://doi.org/10.5061/dryad.nc220.

## Supplemental Information

Supplemental information for this article can be found online at http://dx.doi.org/10.7717/peerj.6142#supplemental-information.

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
