# Peer review of "Fully automated sequence alignment methods are comparable to, and much faster than, traditional methods in large data sets: an example with hepatitis B virus"

_PeerJ, doi:10.7717/peerj.6142_

## Round 0.1 · original submission · Major Revisions

· Academic Editor

Major Revisions

Based on the two reviewer reports, I think that major revisions are necessary before your manuscript can be considered for publication in PeerJ.

Reviewer 2 ·

Basic reporting

no comment

Experimental design

no comment

Validity of the findings

no comment

Additional comments

The authors of 'Fully automated sequence alignment methods ...' curate
and align large numbers of hepatitis B sequences (and fragments), and
use the alignments to estimate phylogenetic trees. They then compare
the accuracies of the tree prediction for a pipeline based on MUSCLE
against one based on UPP/PASTA. I found the paper very 'bitty'. None
of the figures were very insightful, except for Figure 3.B, which is a
piece of art, and in any print journal would be a contender for the
title page. The introduction is very reference heavy for tangential
matter. The standard of written English is good.



[1] The title states that 'fully automated sequence alignment methods
outperform traditional methods'. However, the authors only compare
'fully automated sequence alignment methods' to *one* 'traditional
method' (MUSCLE) not 'methods'.

Action: either amend the title to summarize the work accurately or
include other 'traditional methods'.


[2] How divergent are these sequences? These are all isolates of one
virus type and should actually be relatively easy to align but the
divergence (average percent differences would do) would make that
clear. What would happen if one used GBLOCKS to remove regions of high
alignment instability?

Action: - Quantify similarity/divergence of data
- Comment on using GBLOCKS


[3] On line 139 the authors say recombinants were removed. That makes
sense if one makes trees and compares them as recombinants (between
two or more strains) will have confusing phylogeny. On page 158, the
authors seem to say that recombination implies rearrangement along a
genome which is not the same?

Action: Clarify


[4] The authors pick MUSCLE as their method of choice to compare
against 'fully automated sequence alignment methods'. However, almost
immediately (line 175) they say that 'using MUSCLE to align all 55,353
sequences ... was not feasible'. Why choose MUSCLE, which allegedly
cannot align 55k sequences over CLUSTAL OMEGA, FAMSA, KALIGN or MAFFT,
which can? Also, www.drive5.com/muscle/manual/compromise.html
suggests to use -maxiters 2 for large data-sets. This should make 55k
sequences (more) feasible for MUSCLE. What are and why use custom gap
parameters on line 155 for MUSCLE?

Action: - Explain why UPP/PASTA is compared to MUSCLE and not to any
other aligner/s (or do compare to other aligners).
- Explain why -maxiters 2 is not used for MUSCLE
- Explain 'custom gap parameters' for MUSCLE


[5] In line 162 is the 'branch length' the distance from the root or
distance of a leaf from the nearest branch point?

Action: Clarify


[6] In line 214 five iterations of the NJ QC step may have somewhat
been 'previously described' (I could not recall) but 'NJ QC' was
definitely not formally defined.

Action: define 'NJ QC' before line 214



[7] The authors say that 'sequence alignment strategy did not have a
major effect on tree topology' (line 328). So the main advantage of
'fully automated sequence alignment methods' over 'traditional
methods' would be that they produce results 'in a more reasonable time
frame and repeatable manner' (line 342).

This is difficult to ascertain as (i) no times (or memory
requirements) are quoted, (ii) no 'traditional method', that can
actually align the required number of sequences without human
intervention, is used and (iii) the results are *not* repeatable.

Using PASTA and UPP I have repeatedly (UPP five times, PASTA twice)
aligned the same set of (small in number and short) sequences and
every time obtained a different alignment. In my example the alignment
lengths varied by 7%, the sum-of-pairs scores of one alignment against
another alignment varied from 96% to 97% and the total column scores
from 23% to 40%. For 'repeatable' results I would consistently expect
100%.

If I understand UPP/PASTA correctly, then there is a certain degree of
randomness involved in the assignation of the backbone
sequences. However, I did not see that there was (can be?) any random
seed specified on the command-line. If this is so, then the alignments
have to be repeated several times and the results have to be
somehow combined/averaged.

It is well known that MAFFT L-INS-i is not thread-safe. SATe/PASTA/UPP
use MAFFT L-INS-i (indeed it is my opinion, that much of the quality
of SATe/PASTA/UPP is due to this fact, see below) so a 'repeatable'
analysis should be run such that MAFFT L-INS-i is run single-threaded
(or by setting the MAFFT --threadit flag to 0).

Action: - Quote run-times and memory consumption.
- Guarantee that alignments are repeatable.
- If this cannot be done then repeat alignments and average.


[8] As mentioned above, SATe/PASTA/UPP use MAFFT L-INS-i, L-INS-i is a
high-quality consistency aligner and MUSCLE is not. It would be only
fair to compare like with like and either (i) use MUSCLE as the base
aligner in UPP (PASTA can do this, can UPP?) or (ii) use L-INS-i as
the 'traditional method'. L-INS-i would definitely have problems
aligning 55k sequences, but since in the current analysis sequences
are broken up into 1,269 FASTA files, this should be feasible.

Action: Swap MUSCLE for MAFFT L-INS-i (or vice verse)


[9] In line 77 MAFFT is mentioned but not cited

Action: Cite MAFFT


[10] Year of reference on line 506 (Nguyen et al., 2015) not same as
in text (line 83, Nguyen et al., 2016)

Action: make reference consistent



[11] Fix minor typos


l73-4
- the most important steps in a phylogenetic analyses
+ the most important steps in a phylogenetic analysis
+ the most important steps in phylogenetic analyses


l90
- ... www.ncbi.nlm.nih.gov/genome/viruses/)as a mix of
+ ... www.ncbi.nlm.nih.gov/genome/viruses/) as a mix of


l93/4
- Incorporation of these sequences in phylogenetic analyses are useful,
+ Incorporation of these sequences in phylogenetic analyses is useful,


l98
- a hybrid of alignment methods—an increasingly
+ a hybrid of alignment methods —an increasingly


l170
- PASTA v.1.6.3 (Mirarab 2015)
+ PASTA v.1.6.3 (Mirarab et al., 2015)


l248
- Robinson-Fould (RF) distances (Robinson & Foulds, 1981)
+ Robinson-Foulds (RF) distances (Robinson & Foulds, 1981)


l321
- for most genome/s-region alignment types
+ for most genome/S-region alignment types


l574
- B) Robinson-Fould distances
+ B) Robinson-Foulds distances


l575
- between two phylogenies Darker cells indicate two trees
+ between two phylogenies. Darker cells indicate two trees


l579
- Bottom plots show Robinson-Fould distances.
+ Bottom plots show Robinson-Foulds distances.


Figure 2.B
- Robinson−Fould Distance
+ Robinson−Foulds Distance

Reviewer 3 ·

Basic reporting

This manuscript describes a case study of sequence alignment and phylogenetic analysis of hepatitis B viruses. The authors compared traditional and modern automated approaches for MSA. They resulted in generally similar tree topologies. I have several comments on Experimental design (points 1-4), but I think presentation of the data should be improved first (points 5-9; see the General comments section).

Experimental design

1. It seems to be interesting to use the 10 genotypes for assessing the quality of tree. However, it was difficult for me to follow the discussion on this issue (see also points 7, 9 below). "The test for genotype monophyly showed consistent patterns among the different alignment types (genomes, S-region, total)" (lines 311-312), but in Fig.3A, genotype D, E, F and H show highly inconsistent pattern between "Genomes" and "Total" (and noted in lines 316-320). Which is correct?

2. Related to the above point, it'll be possible to discuss more about why most genotypes are not monophyletic (line 55, line 414). Possible factors would be: (1) convergent evolution, (2) difficulty in tree estimation, (3) gene transfers across different lineage, etc. Factor (3) is consistent with the observation that the "genotype occupancy proportions" differ between different genotypes, in my understanding. If this factor has to be assumed (ie, different regions on a genome have different evolutionary histories), then does it make sense to estimate a single tree from the whole genome?

3. The definition of "genotype occupancy proportion value" should be given as an equation.

4. MAFFT is referred to as a component of traditional analysis (line 77), but it has a function for the "modern" automated approach of this size of data (Katoh & Frith 2012):
% mafft --6merpair --addfragments fragmentary_sequences Aligned_Linearized_Genomes > output
https://mafft.cbrc.jp/alignment/software/addsequences.html#fragments
It'll be better to refer to this method.

Validity of the findings

no comment

Additional comments

I suggest improving/fixing the following issues:

5. The alignment by MUSCLE was seemingly provided as Genome_alignments/Genomes_MuscleAlign.fasta in the raw data, but actually the sequences in this file are unaligned. Another file, Genome_alignments/Genomes_Pasta-Linearized.fasta is aligned, but the length (3423) differs from that (3387) explained in line 273.

6. In line 278, the sequence data set is explained using "Fig.1", but Fig.1 in the main part seems to be a different figure (comparison of trees).

7. In figure 3, different bars/squares have to be distinguished only based on color. Moreover, in panel A, the order of colors differ between the bars (orange, yellow, light green, green, blue, purple, magenta?) and the legend (light green, orange, green, yellow, blue, purple, magenta?). For readers with color blindness, it'll be impossible to interpret this figure. Figure 2 has also a similar problem.

8. I was confused because tables S1 (mentioned in line 306) and S2 (line 313) are not in the Supplemental data in the journal's website, but they are in an external repository (Dryad). Dryad is mentioned only one time in the main text (line 144). It should be stated clearly every time the main text refers to the data in Dryad.

9. It would be helpful for readers if a figure is added to the main part to show a graphical representation of the HBV genome to illustrate the locations of S-region, other genes and the regions associated with the 10 genotypes.

10. A terminological issue: Is it correct to say that most genotypes are paraphyletic (line 361)? From fig.3B, at least genotypes A and B seem to be polyphyletic.

---

## Round 0.2 · Major Revisions

· Academic Editor

Major Revisions

Dear authors,

one of the reviewers has still major points of criticism, he/she even recommended to reject your manuscript. I therefore ask you to take his/her criticism seriously and revise the manuscript accordingly.

Best wishes

Burkhard Morgenstern

Reviewer 2 ·

Basic reporting

The authors of 'Fully automated sequence alignment methods outperform
traditional methods for phylogenetic analysis of hepatitis B viruses'
have responded satisfactorily to almost all the referees'
criticisms. However, their answers to point [1.4] are highly
questionable.

Experimental design

[A] Each of the alignment software that I had suggested in my original
review is able to align the full genome set, most of them on a modest
PC (8 cores, 8Gb RAM), and most of them in under 2 hours, using less
than 2Gb. I had to run MUSCLE on a machine with more memory. The
alignment lengths were comparable to the ones quoted by the authors.

CLUSTAL-OMEGA(v1.2.3) T=101min RAM=1Gb L=3846
FAMSA(v1.2.2) T=14min RAM=1.4Gb L=4631
KALIGN(v2.04) T=108min RAM=200Mb L=4958
MAFFT(def, v7.245) T=23min RAM=600Mb L=4578
MUSCLE(v3.8.31) T=38hours RAM=10Gb L=4909

I was not able to run the full genome data set (or the larger
fragmentary one), using PASTA (v1.6.3 or v1.8.2), neither on a 8Gb nor
a 256Gb nor a 512Gb machine. The Java heap space was exhausted for
Opal.

CLUSTAL-OMEGA, FAMSA, KALIGN and MAFFT-PARTTREE were also able to
align the larger data set containing fragments, however, the alignment
lengths were much longer (8541, 10029, 29503, 83689 respectively).

So there are plenty of aligners that the authors could have used to
add weight to their assertion that 'Fully automated sequence alignment
methods outperform traditional methods for phylogenetic analysis of
hepatitis B viruses' which suggests that 'Fully automated sequence
alignment methods outperform *all* traditional methods for
phylogenetic analysis of hepatitis B viruses'.

It is laudable that the authors did run MAFFT and quote the alignment
length, but it is disappointing that none of the MAFFT alignments are
included in the rest of the analysis.

I do agree that it would not be feasible to use ClustalW, as a quick
scalability analysis (2,5,10,20,50,100 sequences) suggests that the
full genome data set might take more than a week. That is, why I did
not suggest this in my original report. If you do include it
nevertheless, then you should still cite it (Larkin et al, 2007; as
well as CLUSTAL-OMEGA, Sievers et al, 2011; Geneious, Kearse et al,
2012; Sequencher, Gene Codes Corporation, Ann Arbor).


Action: Do include more aligners in your analysis
or clearly state (in the text and in the title) that
you are comparing Pasta/Upp with one method (MUSCLE) only

Validity of the findings

[B] I believe that the inclusion and referencing of software is very
defective.

I could not verify any of the statements concerning Sequencher or
Geneious, as both programs are commercial. It would be better to use
only programs that are free for everyone to use.

The fact that MUSCLE was afforded a dedicated 24-core machine is
pointless, as it is a sequential code. It would be much more useful to
quote the available memory.

The statement about MUSCLE running so much faster under Geneious than
as a stand-alone is intriguing. Since I cannot verify this myself (for
the above mentioned reason) I would need a very good explanation to
let this stand.

For the same reason it is totally unacceptable to run CLUSTAL-OMEGA
through SeaView. Citing the version of SeaView is pointless as (under
Linux) it picks up CLUSTAL-OMEGA from /usr/local/bin/, which is
whatever (unknown) version of CLUSTAL-OMEGA the user has
installed. Also, we cannot know if the (alleged) memory error occurred
in CLUSTAL-OMEGA or SeaView. I did align the genome data set and the
fragment data set with CLUSTAL-OMEGA through SeaView on a PC, and it
did not crash (SeaView v4.5.4, CLUSTAL-OMEGA v1.2.3).

Action: remove all analyses of alignment software
called by secondary programs or give satisfactory
explanations which versions of the aligner is run

---

## Round 0.3 · Minor Revisions

· Academic Editor

Minor Revisions

I am asking the authors to take the comments by reviewer 3 into account and do the required minor revisions.

Reviewer 2 ·

Basic reporting

The authors have addressed all issues raised in a satisfactorily manner.

Experimental design

Experimental design appears to be sound.

Validity of the findings

I believe that the findings, based on the experimental design, are valid.

Additional comments

The manuscript is very much improved.

Reviewer 3 ·

Basic reporting

I suggest revising the title to reflect the observations more simply. If a title says that method A outperforms method B, then readers expect:
* There is a reference that can be regarded to be correct based on a solid basis and
* Estimation by method A is significantly more similar to the reference than estimation by method B.
However, the observations here are:
* The results by methods A and B are not essentially different.
* The results are not fully consistent with an external reference (genotypes classification in this case).
It seems to be difficult to discuss which method is better based on these observations. A possible conclusion would be that automated methods resulted in an apparently reasonable estimation for this specific data. This might be informative since automated methods require less labour.

2. It seems to be a bit misleading to refer to MAFFT in the second sentence in Introduction: 'Traditionally, MSA methods for phylogenetic reconstruction are performed using alignment programs such as MUSCLE or MAFFT followed by manual, “by eye” corrections'. Actually MAFFT has an option designed for adding fragmentary sequences to a backbone, as stated in the last sentence in Introduction.

Experimental design

No comment

Validity of the findings

No comment

---

## Round 0.4 · accepted · Accept

· Academic Editor

Accept

Thanks for making the final minor modifications. I think, the manuscript is publishable now.

Cheers,

Burkhard

#